# Texture Analysis of the Apparent Diffusion Coefficient Focused on Contrast-Enhancing Lesions in Predicting Survival for Bevacizumab-Treated Patients with Recurrent Glioblastoma

**DOI:** 10.3390/cancers15113026

**Published:** 2023-06-01

**Authors:** Antonio Lopez-Rueda, Josep Puig, Santiago Thió-Henestrosa, Javier Luis Moreno-Negrete, Christian Zwanzger, Teresa Pujol, Iban Aldecoa, Estela Pineda, Izaskun Valduvieco, José Juan González, Laura Oleaga

**Affiliations:** 1Department of Radiology (CDI), Hospital Clínic de Barcelona, 08036 Barcelona, Spain; 2Department of Radiology (IDI) and IDIBGI Hospital Universitari de Girona Doctor Josep Trueta, 17190 Girona, Spain; 3Department of Computer Science Applied Mathmatics and Statistics, University of Girona, 17003 Girona, Spain; 4Clínica Iribas-IRM Asunción Paraguay, Asuncion 1430, Paraguay; 5Department of Radiology, Hospital Del Mar, 08003 Barcelona, Spain; 6Department of Anatomical Pathology, Hospital Clínic de Barcelona, 08036 Barcelona, Spain; 7Translational Genomics and Targeted Therapeutics in Solid Tumors Group, Medical Oncology Department, Hospital Clínic de Barcelona, IDIBAPS, University of Barcelona, 08036 Barcelona, Spain; 8Radiotherapy Oncology Service, Hospital Clínic de Barcelona, 08036 Barcelona, Spain; 9Department of Neurosurgery, Laboratory of Experimental Oncological Neurosurgery, Hospital Clínic de Barcelona, 08036 Barcelona, Spain

**Keywords:** glioblastoma, magnetic resonance imaging, biomarkers, diffusion, radiomics, treatment

## Abstract

**Simple Summary:**

After treatment, glioblastoma typically recurs. In some patients with recurrent glioblastoma, bevacizumab improves progression-free survival. The magnetic resonance texture analysis quantifies the macroscopic tissue heterogeneity that is indirectly linked to the microscopic tissue properties. In 33 patients with recurrent glioblastoma who were treated with bevacizumab, we evaluated the predictive value of magnetic resonance texture analysis for survival. Volumes of contrast-enhancing lesions segmented on postcontrast T1-weighted sequences were co-registered with apparent diffusion coefficient maps in order to extract 107 radiomic features. We found that some features derived from texture analysis accurately predicted survival. Identifying pretreatment imaging biomarkers that predict outcomes following bevacizumab therapy for recurrent glioblastoma can be beneficial for selecting patients most likely to benefit from this costly treatment. These promising preliminary results may be a small but significant step toward demonstrating the clinical relevance of radiomic profiles in the treatment of this disease.

**Abstract:**

Purpose: Glioblastoma often recurs after treatment. Bevacizumab increases progression-free survival in some patients with recurrent glioblastoma. Identifying pretreatment predictors of survival can help clinical decision making. Magnetic resonance texture analysis (MRTA) quantifies macroscopic tissue heterogeneity indirectly linked to microscopic tissue properties. We investigated the usefulness of MRTA in predicting survival in patients with recurrent glioblastoma treated with bevacizumab. Methods: We evaluated retrospective longitudinal data from 33 patients (20 men; mean age 56 ± 13 years) who received bevacizumab on the first recurrence of glioblastoma. Volumes of contrast-enhancing lesions segmented on postcontrast T1-weighted sequences were co-registered on apparent diffusion coefficient maps to extract 107 radiomic features. To assess the performance of textural parameters in predicting progression-free survival and overall survival, we used receiver operating characteristic curves, univariate and multivariate regression analysis, and Kaplan–Meier plots. Results: Longer progression-free survival (>6 months) and overall survival (>1 year) were associated with lower values of major axis length (MAL), a lower maximum 2D diameter row (m2Ddr), and higher skewness values. Longer progression-free survival was also associated with higher kurtosis, and longer overall survival with higher elongation values. The model combining MAL, m2Ddr, and skewness best predicted progression-free survival at 6 months (AUC 0.886, 100% sensitivity, 77.8% specificity, 50% PPV, 100% NPV), and the model combining m2Ddr, elongation, and skewness best predicted overall survival (AUC 0.895, 83.3% sensitivity, 85.2% specificity, 55.6% PPV, 95.8% NPV). Conclusions: Our preliminary analyses suggest that in patients with recurrent glioblastoma pretreatment, MRTA helps to predict survival after bevacizumab treatment.

## 1. Introduction

Glioblastoma is the most common and most aggressive primary malignant brain tumor in adults; median survival is 14.5 months, and only 10% of patients survive ≥ 5 years. [1,2]. The current standard of care consists of surgical resection followed by temozolomide in conjunction with radiotherapy [3], but recurrence after treatment is common. Recurrent glioblastoma is often treated with bevacizumab; after bevacizumab treatment, median progression-free survival is 4 months, and median overall survival is 7 months [4,5]. Noninvasive biomarkers that could predict survival in patients with recurrent glioblastoma might help clinicians to select candidates for bevacizumab treatment [6,7].

The growth and progression of glioblastoma require angiogenesis. To promote angiogenesis, glioblastomas secrete various growth factors; among these, vascular endothelial growth factor (VEGF) promotes cerebrovascular permeability and plays a role in tumor progression [8,9]. Bevacizumab, a humanized monoclonal VEGF-blocking antibody, improves progression-free survival in recurrent glioblastoma [10,11,12], although it is ineffective in extending overall survival in newly diagnosed glioblastoma patients [13,14,15,16]. Glioblastomas are highly heterogeneous, and different disease progression rates and degrees of VEGF expression likely influence the response to bevacizumab. 

Although multiparametric magnetic resonance imaging (MRI) findings show significant agreement in terms of morphologic features [17,18], some of which are strongly associated with poor survival, the accuracy of these imaging variables in predicting genetic heterogeneity and survival is still modest. Changes in diffusion-weighted imaging (DWI) have been proposed as a biomarker to predict the response to anti-angiogenesis drugs and apoptosis-inducing treatments [19,20,21,22,23,24,25,26]. A few studies have tried to define metrics for predicting survival and monitoring the response to bevacizumab through histogram analysis of diffusion data; these analyses are based on calculations of apparent diffusion coefficient (ADC) values of the contrast-enhancing lesion within the volume of interest before treatment [20,23,24,25]. Other studies have used voxel-subtraction techniques on functional diffusion maps to predict the local effects of chemotherapy and radiotherapy between two time points [21,26]. Lower values in the pretreatment ADC histogram of contrast-enhancing lesions are associated with poor outcome after bevacizumab treatment for recurrent glioblastoma [18,27]. 

Magnetic resonance texture analysis (MRTA) is an emerging radiomics approach that aims to quantify macroscopic tissue heterogeneity, often imperceptible to the human eye, by analyzing various parameters based on the distribution of pixel values that are indirectly linked to microscopic tissue heterogeneity. Parameters based on geometry (e.g., kurtosis), intensity characteristics (e.g., histograms of pixel distribution), entropy, and other texture-related features are thought to reflect underlying cellular heterogeneity and can be analyzed to help predict survival in patients with recurrent glioblastoma [28]. MRTA metrics have been associated with glioma grade, molecular status, response to treatment, and survival [28,29,30]. Some radiomic profiles have shown their potential to predict methylguanine methyltransferase (MGMT) promoter methylation status and survival in patients with newly diagnosed glioblastoma [31]. However, it remains to be determined whether radiomics is useful in predicting survival in patients with recurrent glioblastoma treated with bevacizumab. This study aims to evaluate the usefulness of MRTA from routinely available MRI sequences in predicting progression-free and overall survival in patients with recurrent glioblastoma treated with bevacizumab. 

## 2. Materials and Methods

### 2.1. Patients

Retrospective analysis of a prospective database was performed. All patients diagnosed with recurrent glioblastoma identified based on clinical and imaging data between December 2009 and December 2018 who were treated with bevacizumab (10 mg/kg every 2 weeks) on first recurrence were eligible for this study. Recurrence was assessed in a multidisciplinary tumor board based on the Response Assessment in Neuro-Oncology (RANO) criteria on the follow-up imaging studies and the clinical status. Patients without valid follow-up MRI studies were excluded. All patients had been treated with temozolomide and radiotherapy following maximal tumor resection. All patients were on corticosteroids at baseline MRI; corticosteroids were discontinued during bevacizumab therapy. We considered only patients with solidly enhancing tumors not consistent with radiation necrosis. Our institution’s ethics committee approved the study protocol, and all patients provided written informed consent. 

### 2.2. MRI Protocol

Patients underwent MRI on a standard clinical 1.5-T system (Intera, Philips Healthcare, Best, The Netherlands) with an eight-channel head coil. The protocol included sagittal 3D T1-weighted imaging, axial fluid-attenuated inversion recovery (FLAIR) imaging, axial diffusion tensor imaging (DTI), axial susceptibility-weighted imaging (SWI), sagittal 3D T2-weighted imaging, first-pass echo-planar dynamic susceptibility-weighted contrast-enhanced (DSC) perfusion imaging with gadobutrol (Gadovist; Bayer Schering Pharma, Berlin, Germany), and sagittal 3D T1-weighted and axial T1-weighted imaging after contrast administration. Parameters for sagittal 3D T1-weighted imaging were repetition time (TR) 12 ms, echo time (TE) 4.6 ms, flip angle (FA) 15°, matrix 256 × 256, section thickness 1 mm, and field of view (FOV) 240 mm. Parameters for FLAIR were TR 9000 ms, TE 164 ms, inversion time (TI) 2500 ms, FA 150°, matrix 256 × 192, section thickness 5 mm, and FOV 240 mm. Parameters for sagittal 3D T2-weighted imaging were TR 3200 ms, TE 402 ms, FA 90°, matrix 256 × 256, section thickness 1 mm, and FOV 240 mm. Parameters for DTI were TR 6900 ms, TE 90 ms (b = 0 and 1000 s/mm^2^), matrix 256 × 256, section thickness 2.4 mm, FOV 240 mm, and number of directions 30. ADC maps were calculated on a voxel-by-voxel basis. Parameters for SWI were TR 26 ms, TE 20 ms, FA 15°, matrix 256 × 192, section thickness 0.75 mm, and FOV 240 mm. Parameters for axial SE T1-weighted images were TR 400 ms, TE 2.6 ms, FA 90°, matrix 448 × 256, section thickness 5 mm, and FOV 240 mm. For perfusion DSC, multislice T2* single-shot echo-planar images were acquired before, during, and after rapid administration of a contrast bolus (twenty-one 5 mm sections without gaps, matrix, 128 × 128, FOV 240 × 240 mm, TR 1550 ms, TE 32 ms, flip angle 90°). Each perfusion series consisted of 50 dynamic acquisitions with the temporal resolution set to 1.8 s during the first pass of a standard dose (0.1 mmol/kg bolus of gadobutrol administered with a power injector at 5 mL/s, followed by a 20 mL bolus of saline at the same rate). To reduce the effect of contrast leakage on calculations of the cerebral blood volume [32], 5 min prior to DSC perfusion acquisitions, a 5 mL bolus of gadobutrol was administered at a rate of 1 mL/s, followed by a 15 mL saline flush. Sagittal 3D T1-weighted imaging and axial T1-weighted imaging were performed after contrast administration with the same parameters previously described.

### 2.3. Image Analysis

Digital imaging and communications in medicine files were transferred to an external computing station for processing. A single neuroradiologist (15 years’ experience) segmented the tumor on gadolinium contrast-enhanced T1-weighted sequences using 3D Slicer software version 4.10.2 [33] (Figure 1). After segmentation, contrast-enhancing lesion volume was mapped to apparent diffusion coefficient maps and quantitative imaging texture features were extracted. To ensure consistency between volumes of interest (VOIs), all depicted VOIs for contrast-enhancing lesions were delineated using the same criteria and were visually validated by the same neuroradiologist. Cystic or necrotic areas were excluded. The resection cavity from the first surgery was also excluded if no sign of contrast enhancement was present. If blood residuals were seen along the border of the resection cavity, the hyperintense pre-contrast T1 volume was subtracted from the post-contrast T1 volume. Textural features were calculated using the SlicerRadiomics module. A total of 107 features pertaining to the First-Order, Shape-Based 3D, Gray Level Co-occurrence Matrix, Gray Level Size Zone Matrix, Gray Level Run Length Matrix, Neighboring Gray Tone Difference Matrix, and Gray Level Dependence Matrix classes were selected from the PyRadiomics lists [34,35].

### 2.4. Statistical Analysis

All variables are expressed as means and standard deviations. Progression-free survival and overall survival were measured from the start of bevacizumab therapy. To identify features that differed significantly between the groups of patients with and without progression-free survival at 6 months and overall survival at 1 year, we used Student’s *t*-tests or non-parametric tests, as appropriate. To determine independent predictors of progression-free survival and of overall survival, we used univariate Cox proportional hazard regression, selecting variables with *p*-values < 0.05 to generate prognostic models and calculating hazard ratios with their corresponding 95% confidence intervals. To determine the optimal cutoffs for these variables, we used receiver operating characteristic curve analysis. To elaborate survival curves, we used the Kaplan–Meier method, including the variables that differed significantly between patients with and without progression-free survival for >6 months and between those with and without overall survival > 1 year. To compensate for the comparison of multiple factors and the small sample size, we applied the Bonferroni correction. Using the log-rank test to evaluate global differences, we also combined these variables to achieve the greatest predictive ability. We used R (Version 3.5.3, The R Foundation, Vienna, Austria) and IBM SPSS (Version 23.0.0.0, IBM Corp., Armonk, NY, USA) for statistical analyses; significance was set at 0.05.

## 3. Results

Patient characteristics. A total of 42 patients met the initial inclusion criteria. Of these, no patients were lost to follow-up, and nine patients were excluded for motion artifacts. Therefore, we retrospectively evaluated 33 patients (20 men; mean age 56 ± 13 years). The median tumor VOI for contrast-enhancing lesions was 29.36 ± 22.68 mL. The mean progression-free survival and overall survival were 5.14 ± 5.49 and 9.23 ± 8.36 mL, respectively. Table 1 summarizes the main characteristics of the study cohort. 

### 3.1. Texture Analysis

Progression-free survival > 6 months was associated with lower major axis length (MAL), lower maximum 2D diameter row (m2Ddr), higher skewness, and higher kurtosis (Table 2). Overall survival > 1 year was associated with lower major axis length (MAL), lower maximum 2D diameter row (m2Ddr), higher skewness, and higher elongation (Table 3). The model including MAL, m2Ddr, and skewness best predicted progression-free survival at 6 months (AUC = 0.886, 100% sensitivity, 77.8% specificity, 50% PPV, 100% NPV) (Table 4). The model including m2Ddr, elongation, and skewness best predicted overall survival at 1 year (AUC = 0.895, 83.3% sensitivity, 85.2% specificity, 55.6% PPV, 95.8% NPV) (Table 5).

### 3.2. Survival Analysis

The Kaplan–Meier plot for progression-free survival shows a statistically significant difference between the group of patients with kurtosis above the cutoff and those with kurtosis below the cutoff (log-rank test, *p* = 0.037; Figure 2). The plot for overall survival shows a trend toward significance for the difference between the groups of patients with m2Ddr above and below the cutoff (log-rank test, *p* = 0.090; Figure 3).

## 4. Discussion

In patients with recurrent glioblastoma, bevacizumab is associated with increased progression-free survival but not with increased overall survival; thus, its cost-effectiveness in this scenario remains uncertain. Identifying pretreatment imaging biomarkers to predict outcomes after bevacizumab therapy for recurrent glioblastoma can be useful for selecting the patients most likely to benefit from this expensive treatment.

To identify imaging parameters associated with the efficacy of bevacizumab, we performed MRTA on DWI from volumes segmented on gadolinium contrast-enhanced T1-weighted sequences in patients with recurrent glioblastoma prior to treatment with bevacizumab. We found that several features’ indexes derived from MRTA accurately predicted survival. Univariate Cox proportional hazards regression found that the best predictors of progression-free survival at 6 months were MAL, m2Ddr, and skewness, and the best predictors of overall survival at one year were m2Ddr, elongation, and skewness. However, in the Kaplan–Meier plots, the only feature that was associated with a significant difference in progression-free survival was kurtosis.

There is growing evidence for the usefulness of MRTA in predicting survival in patients with newly diagnosed glioblastoma [36,37,38,39]. In one recent study, Priya et al. [40] extracted texture features derived from contrast-enhanced T1-weighted images, analyzing the necrotic and contrast-enhancing portions of the tumor after excluding edema. They found that a neural network classifier model combining age and histogram-based first-order textures could differentiate between patients with short (<12 months) and long (>24 months) survival with 70% accuracy. Choi et al. [41] recently found that a model combining radiomic features derived from peritumoral T2 hyperintensity, including texture features, and clinical parameters improved survival prediction in patients with newly diagnosed glioblastoma. In another study, Ingrisch et al. [42] applied radiomic analysis to predict overall survival from the contrast-enhanced lesion segmented on T1-weighted images in a sample of 66 patients with newly diagnosed glioblastoma. Texture parameters were among the most important variables in this model, enabling the authors to differentiate the two patient groups with longer and shorter survival. In line with these studies, Upadhaya et al. [43] also demonstrated that postcontrast imaging texture features provided prognostic value. 

Moreover, an increasing number of studies show that radiomics also provides useful information in recurrent glioblastoma. Grossman et al. [44] performed a radiomic analysis of patients with recurrent glioblastoma treated with bevacizumab, extracting 65 quantitative imaging features from T1 and FLAIR sequences acquired before treatment and 6 weeks after starting treatment. They found that radiomic features helped to predict progression-free survival and overall survival, and textural-imaging heterogeneity was an especially important prognostic factor, independent of volumetric features, age, sex, and Karnofsky performance status. Vils et al. [45] analyzed contrast-enhancing lesion and peritumoral volumes in a cohort of patients with recurrent glioblastoma, extracting 180 radiomic features. Their model predicted O^6^-methylguanine DNA methyltransferase promoter methylation status, but was unable to predict progression-free or overall survival. On the other hand, Huang et al. reported the usefulness of 18fluoromisonidazole (18F-FMISO) PET to evaluate the hypoxia volume in patients with recurrent GBM refractory to bevacizumab, and hypoxia was inversely correlated with OS and PFS [46]. 

The usefulness of diffusion MRI phenotypes in predicting survival in recurrent glioblastoma treated with bevacizumab remains to be determined. Ellingston et al. [47] found an association between lower pretreatment ADC values and lower progression-free survival and overall survival in patients with recurrent glioblastoma treated with bevacizumab. Analyzing data from a randomized, controlled phase III trial comparing the efficacy of bevacizumab with and without VB-111, the same group [48] found that baseline tumor volume and data derived from ADC histogram analysis were predictive biomarkers of overall survival. As in our study, the histograms were generated from ADC values extracted from contrast-enhancing regions in T1-weighted images. For recurrent glioblastoma with a large tumor burden, diffusion MR phenotypes can predict overall survival in patients treated with bevacizumab or surgical resection [49]. Finally, a recent systematic review and meta-analysis evaluating the predictive value of the mean ADC value of the lower Gaussian curve (ADCL) derived from bi-Gaussian curve-fitting histogram analysis in patients with recurrent glioblastoma found that low ADCL was associated with decreased progression-free survival and overall survival after bevacizumab treatment [28]. 

We did not analyze the relationship between ADC values and survival together with data from histologic or gene expression studies [50]. Thus, it is impossible to know the extent to which low ADC values in the recurrent glioblastomas that respond poorly to bevacizumab are related to hypoxia or hypercellularity. We used the texture characteristics skewness and kurtosis to measure the distribution of ADC values around the mean. The values of these parameters reflect the shape of a histogram: skewness measures the symmetry of the distribution, and kurtosis measures the weight of its tails, thus depending on the frequency with which outliers occur. In a normal distribution, skewness equals zero and kurtosis equals three. Skewness will be positive if more data are concentrated on the left of the histogram and negative if more data are concentrated on the right. Values of kurtosis greater than 3 indicate that the sample has a greater proportion of patients in the tails compared to a normal distribution [51].

Skewness and kurtosis are indicators of tumor heterogeneity and can provide valuable information for differential tumor diagnosis [52]. In Figure 4, we provide a hypothetical biological explanation for our main results. The black line represents a hypothetical normal distribution curve (skewness = 0 and kurtosis = 3) where patients would be equally distributed according to their ADC values in contrast-enhancing lesions. The blue line represents the distribution of patients with longer progression-free survival (>6 months) according to their ADC values in contrast-enhancing lesions, and the red line represents the distribution of the patients with shorter progression-free survival (≤6 months) according to the same parameter. The box below the graph depicts the relationship between ADC and cellularity, showing how low ADC values in a region of interest could represent high tumor cellularity, and higher ADC values could represent areas of low tumor cellularity. The curve for patients with longer progression-free survival corresponds to fewer areas of low ADC values in enhancing tumors (skewness 1.5) (in the box delimited by the broken red line) and more areas of enhancing tumors with ADC values that are not as low (in the box delimited by the broken green line). In other words, patients with longer progression-free survival (blue curve) would have fewer high-cellularity areas and more low-cellularity areas in the tumor than patients with shorter progression-free survival (red curve).

Some limitations of our study merit comment. Our small sample limits the generalizability of our findings; additional research in larger cohorts and external validation are necessary to confirm the usefulness of our model. Another limitation of our study is the lack of data from histologic or gene expression studies and its relationship with patient survival. Various aspects of the biology of glioblastoma progression and treatment-induced alterations affect survival, and this complexity can be hard to capture in models. The molecular profile of glioblastomas at initial detection is different from that of recurrent glioblastomas, indicating a change in tumor biology [53]. Moreover, after primary treatment, varying amounts of scar tissue, resection cavities, and the extent of resection may also affect the imaging appearance of a recurrent tumor; moreover, the tumor volume at recurrence is often very small, resulting in fewer tumor data for calculations compared to the initial diagnosis. 

Most quantitative radiomics research has focused on newly diagnosed glioblastomas, and very few studies have used radiomic approaches to study recurrent glioblastomas [31,54,55,56]. Our study aimed to propose models to predict survival after bevacizumab for recurrent glioblastoma. These promising preliminary results may be a small but significant step toward demonstrating the clinical relevance of radiomic profiles in this disease.

## 5. Conclusions

In conclusion, for enhancing lesions segmented on post-contrast T1-weighted sequences and mapped to ADC maps, the textural features of diffusion kurtosis may help to predict progression-free survival in patients with recurrent glioblastoma before initiating treatment with bevacizumab.

## Figures and Tables

**Figure 1 cancers-15-03026-f001:**
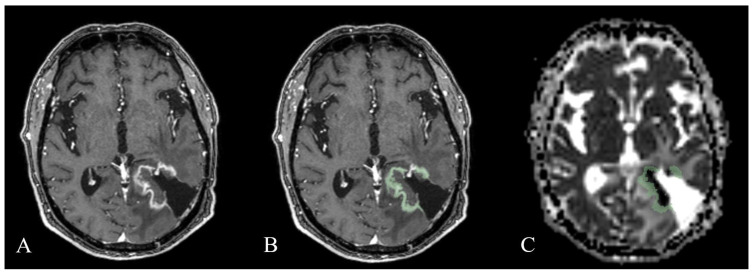
Tumor segmentation. (**A**,**B**) Contrast-enhancing lesions were segmented from T1-weighted images (green). After segmentation, contrast-enhancing lesion volume was mapped to apparent diffusion coefficient maps. (**C**) Quantitative imaging texture features were extracted from volumes of interest.

**Figure 2 cancers-15-03026-f002:**
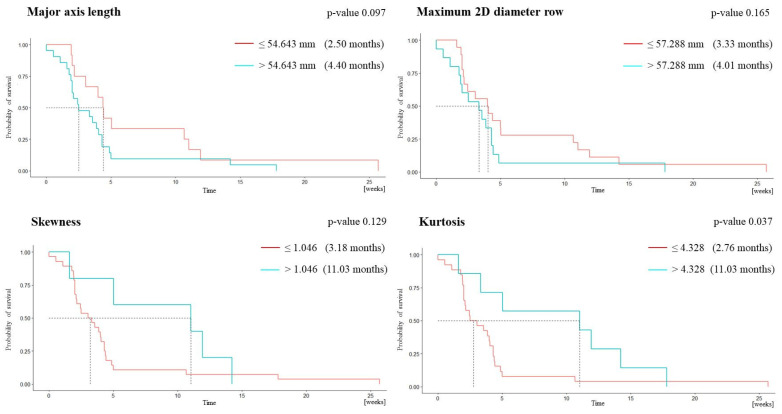
Kaplan–Meier plots for progression-free survival.

**Figure 3 cancers-15-03026-f003:**
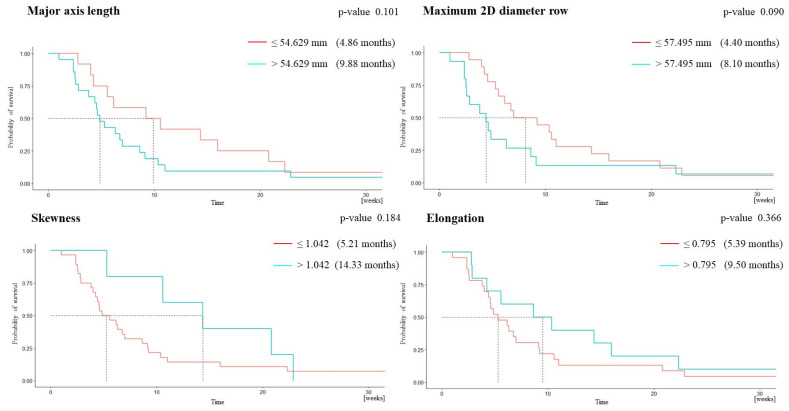
Kaplan–Meier plots for overall survival.

**Figure 4 cancers-15-03026-f004:**
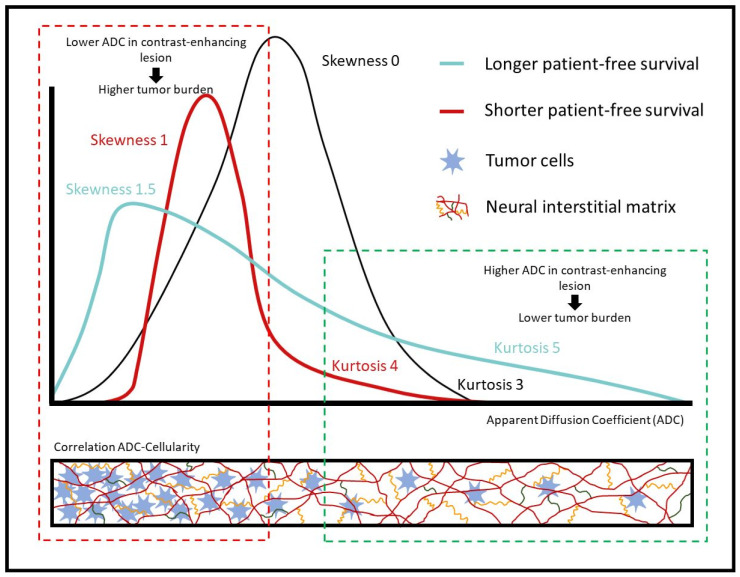
Hypothetical biological explanation for the ADC-texture analysis of contrast-enhancing lesions for predicting survival in patients with recurrent glioblastoma after treatment with bevacizumab.

**Table 1 cancers-15-03026-t001:** Characteristics of patients with recurrent glioblastoma included in the study (n = 33).

Age (years) (mean, SD)	56; 13
Female (n, %)	13; 39.39%
Contrast-enhancing lesion volume (mL) (mean, SD)	29.36; 22.68
PFS (months) (mean, SD)	5.14; 5.49
≤6 months group	2.89; 1.41
>6 months group	15.23; 5.77
OS (months) (mean, SD)	9.23; 8.36
≤1 year group	5.49; 2.86
>1 year group	23.12; 7.30

**Table 2 cancers-15-03026-t002:** Features associated with progression-free survival beyond 6 months.

Feature	Whole Cohort (n = 33)	PFS ≤ 6 Months (n = 27)	PFS > 6 Months (n = 6)	*p*-Value
MAL	63.11 ± 21.11	66.46 ± 21.31	48.01 ± 12.54	0.027
m2Ddr	56.51 ± 17.82	59.07 ± 18.28	45 ± 10.03	0.021
Skewness	0.68 ± 0.46	0.60 ± 0.42	1.06 ± 0.48	0.021
Kurtosis	3.66 ± 1.89	3.36 ± 1.70	4.98 ± 2.30	0.035

PFS, progression-free survival; MAL, major axis length; m2Ddr, maximum 2D diameter row.

**Table 3 cancers-15-03026-t003:** List of significant features associated with overall survival at 1 year.

Feature	Whole Cohort (n = 33)	OS ≤ 1 Year (n = 26)	OS > 1 Year (n = 7)	*p*-Value
MAL	63.11 ± 21.11	66.99 ± 21.55	48.68 ± 11.58	0.021
Elongation	0.66 ± 0.17	0.63 ± 0.18	0.77 ± 0.12	0.027
m2Ddr	56.51 ± 17.82	59.12 ± 18.64	46.83 ± 10.36	0.034
Skewness	0.68 ± 0.46	0.60 ± 0.42	0.98 ± 0.48	0.043

OS, overall survival; MAL, major axis length; m2Ddr, maximum 2D diameter row.

**Table 4 cancers-15-03026-t004:** Survival prediction: summary of class performance for progression-free survival *.

Univariate Analysis						
Feature	AUC	Sensitivity	Specificity	PPV	NPV	Cut-Off Value
MAL	0.790	0.667	0.704	0.333	0.905	54.643
m2Ddr	0.747	0.833	0.519	0.278	0.933	57.288
Skewness	0.802	0.500	0.926	0.6	0.893	1.046
Kurtosis	0.769	0.667	0.889	0.571	0.923	4.328
Bivariate Analysis						
MAL m2Ddr	0.880	0.833	0.815	0.5	0.957	96.436 86.200
MAL Skewness	0.880	1.000	0.741	0.462	1.000	96.436 0.653
Trivariate Analysis						
MAL m2Ddr Skewness	0.886	1.000	0.778	0.5	1.000	

(*) Only the best-performing bivariate and trivariate models are presented. AUC, area under the receiver operating curve; PPV, positive predictive value; NPV, negative predictive value; MAL, major axis length; m2Ddr, maximum 2D diameter row.

**Table 5 cancers-15-03026-t005:** Survival prediction: summary of class performance for overall survival.

Univariate Analysis						
	AUC	Sensibility	Specificity	PPV	NPV	Cut-Off Value
MAL	0.788	0.714	0.731	0.417	0.905	54.629
m2Ddr	0.712	0.714	0.5	0.278	0.867	57.495
Skewness	0.742	0.429	0.923	0.6	0.857	1.042
Elongation	0.728	0.571	0.769	0.4	0.87	0.795
Bivariate Analysis						
MAL Skewness	0.880	1.000	0.741	0.462	1	96.436 0.653
Trivariate Analysis						
m2Ddr Elongation Skewness	0.895	0.833	0.852	0.556	0.958	

Only the best-performing bivariate and trivariate models are presented. AUC, area under the receiver operating curve; PPV, positive predictive value; NPV, negative predictive value; MAL, major axis length; m2Ddr, maximum 2D diameter row.

## Data Availability

The data presented in this study are available in this article.

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
