# Peer review of "Texture Analysis of the Apparent Diffusion Coefficient Focused on Contrast-Enhancing Lesions in Predicting Survival for Bevacizumab-Treated Patients with Recurrent Glioblastoma"

_cancers, 2023, doi:10.3390/cancers15113026_

Round 1

Reviewer 1 Report

This retrospective single-center study involved 33 glioblastoma patients at first recurrence treated with bevacizumab. The contrast-enhanced (Gd-T1) and the diffusion-weighted MRI of the patients before receiving the antiangiogenic therapy were used by the authors to apply texture analysis in the DWI data (ADC maps) at the corresponding regions of contrast-enhancing recurrent tumor, to identify the potential of the extracted diffusion textural features in predicting bevacizumab efficacy by prolonging progression-free and/or overall patient survival.

Beneath is a list of points that need to be further addressed by the authors:

1. Materials and Methods – Patients (page 2): The authors state as late as in the Results (top of page 4) that this study was retrospective. Still, the clear statement about the retrospective nature of the study must be made in the Materials and Methods section rather than in the Results. Furthermore, the authors state –at the end of the “Patients” paragraph– that all enrolled patients provided written informed consent. How patient consent was attainable in retrospect, considering that the enrollment period spanned 9 years (2009-2018) and the disease under consideration bears a dismal prognosis (median OS <1 year)?

2. Materials and Methods – MRI protocol (pages 2–3): The matrix resolution is expressed in pixels rather than in mm.

3. Materials and Methods – Image analysis (page 3): For reasons of clarity, it should be added also in the paragraph text that the quantitative textural radiomics features were extracted from the ADC maps, with the Gd-T1 images serving for lesion segmentation and VOI definition. (Currently, this is stated clearly only in the legend of the corresponding Figure 1).

4. Materials and Methods – Image analysis (page 3): Regardless of how experienced a neuroradiologist might be, relying on a single reader’s sole image analysis may be regarded as a shortcoming of the study. An optimized analysis approach would involve analyzing each patient’s MRI exam twice to assess intra-reader repeatability, or –better still– an analysis by two separate experienced readers. The study did not address at all the level of the extracted radiomics features’ robustness to the intra-/inter-reader image segmentation variability.

5. Statistical analysis (top of page 4): The correct sequence of the software and vendor is: IBM SPSS (Version 23.0.0.0, IBM Corp., Armonk, NY).

6. Legends of Tables 2 & 3 (page 4): Please revise/improve the use of English.

7. Table 3 – Header of column #4 (page 4): It is OS> 1 year.

8. Conclusion (top of page 10): In its current form, it is very generalized and does not reflect the specific findings of the study. (The same also applies to the Abstract Conclusion.) A more relevant statement would read like this: “In conclusion, for enhancing lesions segmented on post-contrast T1-weighted sequence and mapped to the ADC map, the textural feature of diffusion kurtosis may help to predict progression-free survival in patients with recurrent glioblastoma before initiating treatment with bevacizumab.”

9. References (page 11): The two last references (#52 and #53) are not cited in the manuscript text.

Author Response

Reviewer 1

This retrospective single-center study involved 33 glioblastoma patients at first recurrence treated with bevacizumab. The contrast-enhanced (Gd-T1) and the diffusion-weighted MRI of the patients before receiving the antiangiogenic therapy were used by the authors to apply texture analysis in the DWI data (ADC maps) at the corresponding regions of contrast-enhancing recurrent tumor, to identify the potential of the extracted diffusion textural features in predicting bevacizumab efficacy by prolonging progression-free and/or overall patient survival.

Beneath is a list of points that need to be further addressed by the authors:

  1. Materials and Methods – Patients (page 2): The authors state as late as in the Results (top of page 4) that this study was retrospective. Still, the clear statement about the retrospective nature of the study must be made in the Materials and Methods section rather than in the Results. Furthermore, the authors state –at the end of the “Patients” paragraph– that all enrolled patients provided written informed consent. How patient consent was attainable in retrospect, considering that the enrollment period spanned 9 years (2009-2018) and the disease under consideration bears a dismal prognosis (median OS <1 year)?

Response: This is a retrospective analysis of a prospective database, where the patients provided written informed consent to use their data for research purposes.

Added to the manuscript, in Materials and Method section.

  1. Materials and Methods – MRI protocol (pages 2–3): The matrix resolution is expressed in pixels rather than in mm.

Response: You are absolutely right. We are sorry for the mistake, we have change it accordingly. Changed in the manuscript.

  1. Materials and Methods – Image analysis (page 3): For reasons of clarity, it should be added also in the paragraph text that the quantitative textural radiomics features were extracted from the ADC maps, with the Gd-T1 images serving for lesion segmentation and VOI definition. (Currently, this is stated clearly only in the legend of the corresponding Figure 1).

Response: Added to the manuscript.

  1. Materials and Methods – Image analysis (page 3): Regardless of how experienced a neuroradiologist might be, relying on a single reader’s sole image analysis may be regarded as a shortcoming of the study. An optimized analysis approach would involve analyzing each patient’s MRI exam twice to assess intra-reader repeatability, or –better still– an analysis by two separate experienced readers. The study did not address at all the level of the extracted radiomics features’ robustness to the intra-/inter-reader image segmentation variability.

Response: We relied on one experienced reader only due to the easy segmentation of the enhancing lesions. We did not perform a second segmentation because the segmentations of the first 10 patients were reviewed by a second neuroradiologist and they coincided in the segmentation maps.

  1. Statistical analysis (top of page 4): The correct sequence of the software and vendor is: IBM SPSS (Version 23.0.0.0, IBM Corp., Armonk, NY).

Response: Added to the manuscript.

  1. Legends of Tables 2 & 3 (page 4): Please revise/improve the use of English.

Response: Some typos have been corrected in these tables.

  1. Table 3 – Header of column #4 (page 4): It is OS> 1 year.

Response: Added to the manuscript.

  1. Conclusion (top of page 10): In its current form, it is very generalized and does not reflect the specific findings of the study. (The same also applies to the Abstract Conclusion.) A more relevant statement would read like this: “In conclusion, for enhancing lesions segmented on post-contrast T1-weighted sequence and mapped to the ADC map, the textural feature of diffusion kurtosis may help to predict progression-free survival in patients with recurrent glioblastoma before initiating treatment with bevacizumab.”

Response: The reviewer is right. Added to the manuscript

  1. References (page 11): The two last references (#52 and #53) are not cited in the manuscript text.

Response: Thanks for this notification. We have included these two references at the end of the discussion.

Reviewer 2 Report

Reviewer’s comments

Manuscript number: cancers-2207479

Title: Texture analysis of apparent diffusion coefficient focused on contrast-enhancing lesion in predicting survival for bevacizumab-treated patients with recurrent glioblastoma

Summary

The authors analyzed the usefulness of Magnetic resonance texture analysis (MRTA) in predicting outcomes of patients with recurrent glioblastoma treated with bevacizumab. MRTA is a radiomics approach that aims to quantify macroscopic tissue heterogeneity by analyzing various parameters based on the distribution of pixel values. MRTA have been associated with glioma grade, molecular status, treatment response, and survival. In this study, they suggested that MRTA helps predict survival in patients with recurrent glioblastoma treated with bevacizumab.

Specific comments:

Overall, I think it's a well-written article. First of all, congratulations on your successful research results. However, in order to accept the research results, the following problems need to be corrected.

1.       Were the patients in this study diagnosed with recurrent GBMs histologically?

2.       If the diagnosis was made by MR imaging, it seems necessary to describe the diagnostic criteria through other MR sequences, including CBV, DWI, and/or SWI.

3.       Compared according to recurrence and survival time in table 2 and 3, it seems necessary to describe the molecular biological prognostic factors, including IDH mutation and MGMT methylation, of the tumor.

4.       In Materials and Methods section, the authors mentioned “We considered only patients with progressive non-enhancing or solidly enhancing tumors not consistent with radiation necrosis”. And the authors demonstrated tumor segmentation in figure 1. It would be helpful to add a case of progressive non-enhancing tumors.

Overall impression

I would recommend resubmission of the manuscript including considerable revisions.

Thank you very much.

Author Response

Reviewer 2

Summary

The authors analyzed the usefulness of Magnetic resonance texture analysis (MRTA) in predicting outcomes of patients with recurrent glioblastoma treated with bevacizumab. MRTA is a radiomics approach that aims to quantify macroscopic tissue heterogeneity by analyzing various parameters based on the distribution of pixel values. MRTA have been associated with glioma grade, molecular status, treatment response, and survival. In this study, they suggested that MRTA helps predict survival in patients with recurrent glioblastoma treated with bevacizumab.

Specific comments:

Overall, I think it's a well-written article. First of all, congratulations on your successful research results. However, in order to accept the research results, the following problems need to be corrected.

  1. Were the patients in this study diagnosed with recurrent GBMs histologically?

Response: None of the patients had a second surgery to confirm histologically the recurrent GBM status, we relied on the imaging follow up studies and the clinical evaluation.

  1. If the diagnosis was made by MR imaging, it seems necessary to describe the diagnostic criteria through other MR sequences, including CBV, DWI, and/or SWI.

Response: Recurrence was assessed in a multidisciplinary tumor board based on the Response Assessment in Neuro-Oncology (RANO) criteria on the follow up imaging studies and the clinical status. This remark has been added to the manuscript (Material and Methods section).

  1. Compared according to recurrence and survival time in table 2 and 3, it seems necessary to describe the molecular biological prognostic factors, including IDH mutation and MGMT methylation, of the tumor.

Response: All tumors were IDH wild type. We did not have MGMT status, that is the reason why these data are not included in the table.

  1. In Materials and Methods section, the authors mentioned “We considered only patients with progressive non-enhancing or solidly enhancing tumors not consistent with radiation necrosis”. And the authors demonstrated tumor segmentation in figure 1. It would be helpful to add a case of progressive non-enhancing tumors.

Response: This was a drafting error. We have changed this correct sentence in the manuscript: “We considered only patients with solidly enhancing tumors not consistent with radiation necrosis”.

Overall impression

I would recommend resubmission of the manuscript including considerable revisions.

Thank you very much.

Reviewer 3 Report

This paper seems very interesting, look at these points to improve it:

- "A total of 42 patients met the initial inclusion criteria. Of these, no patients were lost to follow-up, and 9 patients were excluded for artifacts" What do the authors mean by "artifacts"? 

- "Univariate Cox proportional hazards regression found the best predictors of progression-free survival at 6 months were MAL, m2Ddr, and skewness, and the best predictors of overall survival at one year were m2Ddr, elongation, and skewness". Please, revise this sentence better to highlight the role of m2Ddr in PFS.

- In the discussion section: "There is growing evidence for the usefulness of MRTA in predicting survival in patients with newly diagnosed glioblastoma" Some missing references should be considered:  --  PMCID: PMC9223226  doi: 10.3390/genes13061054   --  PMID: 35608757  doi: 10.1007/s11547-022-01502-8  --   PMID: 32448612  doi: 10.1016/bs.irn.2020.03.009  --  PMID: 19381441  doi: 10.1007/s11060-009-9897-1

- "We did not analyze the relationship between ADC values and survival together with data from histologic or gene expression studies" Is this a "limitation of the paper"? State it openly in a subparagraph.

- "Moreover, an increasing number of studies show that radiomics also provides useful information in recurrent glioblastoma. Grossman et al. [39] performed a radiomic analysis of patients with recurrent glioblastoma treated with bevacizumab..." What about the role of PET in recurrent GBM treated with bevacizumab? Refs. PMCID: PMC8027395 DOI: 10.1038/s41598-021-84331-5

- Figure 4 seems to not reflect the whole paper. Is it necessary ?

- Conclusion is too short. Try to improve it. What this paper add new to the literature?

Author Response

Reviewer 3

Comments and Suggestions for Authors

This paper seems very interesting, look at these points to improve it:

- "A total of 42 patients met the initial inclusion criteria. Of these, no patients were lost to follow-up, and 9 patients were excluded for artifacts" What do the authors mean by "artifacts"?

Response: Motion artifacts making impossible the segmentation and the extraction of the data. Added to the manuscript (Results section – Patients characteristics).

- "Univariate Cox proportional hazards regression found the best predictors of progression-free survival at 6 months were MAL, m2Ddr, and skewness, and the best predictors of overall survival at one year were m2Ddr, elongation, and skewness". Please, revise this sentence better to highlight the role of m2Ddr in PFS.

Response: We would rather not give greater emphasis to m2Ddr than other indexes because we interpret that the three linked markers  provide the best predicted model (AUC=0.886 for PFS).

- In the discussion section: "There is growing evidence for the usefulness of MRTA in predicting survival in patients with newly diagnosed glioblastoma" Some missing references should be considered:  --  PMCID: PMC9223226  doi: 10.3390/genes13061054   --  PMID: 35608757  doi: 10.1007/s11547-022-01502-8  --   PMID: 32448612  doi: 10.1016/bs.irn.2020.03.009  --  PMID: 19381441  doi: 10.1007/s11060-009-9897-1

Response: All these references have been inserted in the discussion. All references have been adjusted in the rest of the text.

- "We did not analyze the relationship between ADC values and survival together with data from histologic or gene expression studies" Is this a "limitation of the paper"? State it openly in a subparagraph.

Response: Added to the manuscript.

- "Moreover, an increasing number of studies show that radiomics also provides useful information in recurrent glioblastoma. Grossman et al. [39] performed a radiomic analysis of patients with recurrent glioblastoma treated with bevacizumab..." What about the role of PET in recurrent GBM treated with bevacizumab? Refs. PMCID: PMC8027395 DOI: 10.1038/s41598-021-84331-5F-

Response: Added to the manuscript.

- Figure 4 seems to not reflect the whole paper. Is it necessary ?

Response: We believe that providing a hypothetical biological explanation may be interesting to understand the meaning of biomarkers.

- Conclusion is too short. Try to improve it. What this paper add new to the literature?

Response: Added to the manuscript.

Round 2

Reviewer 2 Report

Dear authors

Thank you very much for your responses.

I’ve read the responses for reviewer’s comments.

Revisions are generally satisfactory.

I feel that this manuscript contains valuable information worthy of publication in Cancers.

Thank you.

Reviewer 3 Report

Authors solved all my criticisms